# Quantifying Lithium Ion Exchange in Solid Electrolyte Interphase (SEI) on Graphite Anode Surfaces

**Janet S. Ho** [1,*], **Zihua Zhu** [2,*], **Philip Stallworth** [3], **Steve G. Greenbaum** [3,*] , **Sheng S. Zhang** [1] and **Kang Xu** [1]

1    The DEVCOM Army Research Laboratory, Energy Sciences Division, Sensors and Electron Devices Directorate, Adelphi, MD 20783, USA; shengshui.zhang.civ@army.mil (S.S.Z.); conrad.k.xu.civ@army.mil (K.X.)

2    W. R. Wiley Environmental Molecular Sciences Laboratory, Pacific Northwest National Laboratory, Richland, WA 99354, USA

3    Department of Physics & Astronomy, Hunter College of CUNY, New York, NY 10065, USA; pstallwo@hunter.cuny.edu

\*    Correspondence: janet.ho2.civ@army.mil (J.S.H.); zihua.zhu@pnnl.gov (Z.Z.); steve.greenbaum@hunter.cuny.edu (S.G.G.); Tel.: +1-(301)394-0051 (J.S.H.); +1-(509)371-6240 (Z.Z.); +1-(212)772-4973 (S.G.G.)

**Abstract:** Solid Electrolyte Interphase (SEI) has been identified as the most important and least understood component in lithium-ion batteries. Despite extensive studies in the past two decades, a few mysteries remain: what is the chemical form of and degree of mobility of $Li^+$ in the interphase? What fraction of $Li^+$ is permanently immobilized in the SEI, while the rest are still able to participate in the cell reactions via the ion-exchange process with $Li^+$ in the electrolyte? This study attempted to answer, in part, these questions by using $^6Li$ and $^7Li$-isotopes to label SEIs and electrolytes, and then quantifying the distribution of permanently immobilized and ion-exchangeable $Li^+$ with solid-state NMR and ToF-SIMS. The results showed that the majority of $Li^+$ were exchanged after one SEI formation cycle, and a complete exchange after 25 cycles. Ion exchange by diffusion based on concentration gradient in the absence of applied potential also occurred simultaneously. This knowledge will provide a foundation for not only understanding but also designing better SEIs for future battery chemistries.

**Keywords:** solid electrolyte interphase; graphite anodes; lithium isotopes; capacity loss; time-of-flight secondary-ion mass spectroscopy; solid-state NMR





## 1. Introduction

Graphite is widely used as an anode material in rechargeable lithium ion batteries for a number of reasons. The realization that graphite, being an anode intercalation host, can form intercalation compounds with lithium ions, gave rise to the present-day rechargeable lithium ion batteries (LIBs) that eliminated the safety hazard of lithium dendrite formation in lithium metal batteries. Of the intercalation compounds formed with the carbonaceous anode, $LiC_6$ (i.e., $C_6 + xLi^+ + xe^- \rightleftharpoons Li_xC_6$) is the most lithium-enriched form, and has similar chemical reactivity to lithium metal, thus rendering the anode potential close to that of lithium metal with little energetic penalty. The attraction of this anode material is further highlighted by the low cost of carbon materials, by its stability, and by its environmental friendliness [1–4].

Similar to lithium, the graphite anode surface, when operating in non-aqueous electrolytes, also forms a passivation layer, termed Solid Electrolyte Interphase (SEI), which enables LIBs to operate reversibly if the SEI is well-formed [5,6]. The quality of an SEI has significant impact on the performance of LIBs, such as cycle life and stability, and depends on not only the electrolyte composition but also the type and morphology of carbon, electrochemical conditions, and temperature during formation. An ideal SEI should

maintain high Li ion conductivity but block solvent molecules, and electrons, to prevent any further unwanted decomposition of the electrolyte or solvent co-intercalation that leads to exfoliation of the graphite layers. Furthermore, it needs to have high mechanical strength to withstand stress caused by expansion and contraction of the graphite layers during charging and discharging, respectively. Stability over a wide range of operating temperatures and voltages is another key property [1–4,7].

Because of the importance of a well-formed SEI to the battery performance, extensive research has been done to provide understanding of the formation mechanism, chemical composition, morphology, and physical properties. Several review articles and books have been dedicated to the subject [3,7–10]. While a few different models have been postulated for the formation mechanism [11–14], the general agreement about the SEI structure is that it consists of dual layers. The inner layer is a dense composite of insoluble inorganic salts, such as lithium fluoride (LiF) and lithium oxide ($Li_2O$). The outer layer is an amorphous mixture of insoluble inorganic and organic compounds like lithium carbonate ($Li_2CO_3$), lithium alkyl carbonate, alkoxides, plus non-conducting polymers, from reduction of the solvent molecules; these are identified using techniques such as X-ray photoelectron spectroscopy (XPS), Fourier-transformed infrared spectroscopy (FTIR), and mass spectroscopy [10,15–21]. For a tri-fluoro-methane-sulfonyl-imide ($TFSI^-$) and carbonate-based electrolyte, the composition of SEI was found to be mostly $Li_2CO_3$ and lithium alkyl carbonate (40–70%) with a small amount of LiF (<5%) [16].

The formation of these SEI components consumes Li ions from the 'energy storage inventory'. The amount of Li ions consumed was studied by Diehl, et al. [22], using a $^6Li$-enriched electrolyte as a tracer and laser ablation inductively coupled plasma mass spectrometer to measure Li abundance. They found that the $^6Li$ abundance in the electrolyte decreased from 93.3% to 49.6%, while that in the delithiated graphite anode increased to 43.7%, and the cathode rose from 8.4% to 31% after one formation cycle. No significant changes in the $^6Li$ abundance were found upon further cycling.

The aim of this study was to provide further knowledge on the nature of the Li ions in the SEI on delithiated graphite anodes, by studying the amount of Li isotope exchange in the SEI after switching Li isotope-enriched electrolytes and foils. The isotope exchange was performed electrochemically through charge-discharge cycles. The other approach was by soaking the delithiated graphite anode in the other isotope-enriched electrolyte to induce ion exchange through concentration gradient diffusion. However, this approach was not performed, after the recent work by Berthault, et al., [23] who carried out similar studies, came to our attention. Quantitative measurements of isotopic abundance were conducted using time-of-flight secondary ion mass spectroscopy (ToF-SIMS) for depth profiling and solid-state NMR for bulk analysis. The impact on capacity loss was also studied.

## 2. Results

This study compared SEI formed from one formation cycle, and from 25 cycles, in a graphite anode half-cell configuration. An SEI was first formed using $^7Li$-enriched electrolyte and foil, after which, the solvent-rinsed delithiated graphite anode was reassembled and cycled again using the same number of cycles as the original SEI but with $^6Li$-enriched electrolyte and foil. Controls for each of the isotopes without undergoing isotopic exchange were included as references.

### 2.1. Depth Profiling by ToF-SIMS

ToF-SIMS analysis of the $^6Li$ composition in various delithiated graphite anodes are shown in Figure 1. The $^7Li$-enriched control (Figure 1a) and $^6Li$-enriched control (Figure 1b) showed 0% and 96%, respectively, which agreed with the specifications provided by the Li supplier, while for the samples that had undergone isotope exchange, about 90% of the $^7Li$ in the original SEI was replaced by $^6Li$ after one formation cycle (Figure 1c). Further cycling (25 cycles) increased the amount of $^6Li$ close to that of the $^6Li$-enriched control (Figure 1d). These results show that the majority of the exchange happened in the first

lithiation/delithiation cycle. Also shown in Figure 1c,d are the intensity ratio of $^6$Li to carbon (C), which exhibited a maximum at around 20 to 40 s of sputtering. Given that the SEI was composed of mostly $Li_2CO_3$ and lithium alkyl carbonate, the results suggested that the SEI layer was within about 100 s of sputtering time. While an equivalent sputter rate of 0.5 nm/s, based on amorphous carbon, was used in this study, the relatively large scan area (350 × 350 μm$^2$) and the rough surface of the graphite electrode made SEI thickness determination unreliable. These findings are similar to those reported by Berthault, et al., [23].

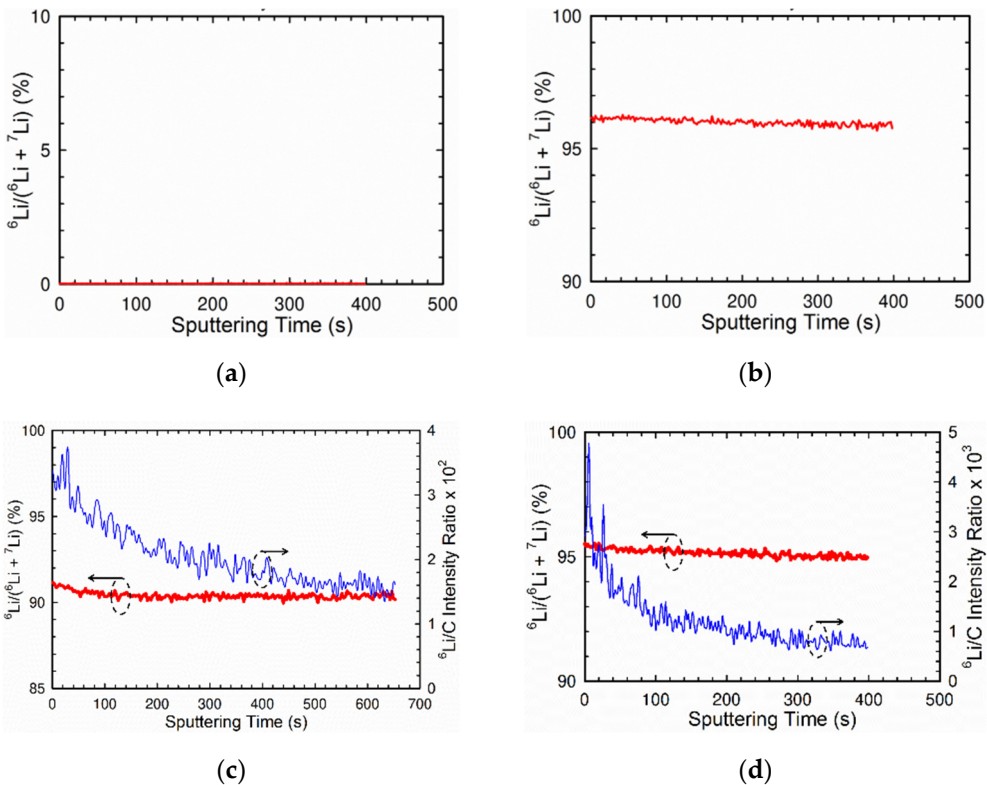

**Figure 1.** $^6$Li composition and $^6$Li to carbon (C) intensity ratio as a function of sputtering time for delithiated graphite anodes after different numbers of SEI formation cycles: (**a**) $^7$Li-enriched control; (**b**) $^6$Li-enriched control. (**c**) $^7$Li-enriched SEI exchanged with $^6$Li-enriched electrolyte, one formation cycle; (**d**) $^7$Li-enriched SEI exchanged with $^6$Li-enriched electrolyte, 25 formation cycles.

### 2.2. Bulk Analysis by Solid State NMR

NMR was another technique employed to determine the isotope composition in the SEI after exchange. The SEI was formed after one formation cycle. The $^7$Li and $^6$Li spectra for $^7$Li-control, $^6$Li-control, and the sample after isotope exchange (labelled as $^7$Li-to-$^6$Li) are shown in Figure 2. The observed $^7$Li signals for $^7$Li-control and Sample $^7$Li-to-$^6$Li (Figure 2a) appear to be symmetric and single peaked, although the spinning sidebands are not resolved. The peak positions are near 0 ppm, and no resolvable $LiC_6$ features at +48 ppm are present [24]. Line shapes and widths (~10kHz) are consistent with an assignment of SEI-distributed $Li^+$ sites including LiF, electrolyte decomposition products, etc. The spectra were normalized and integrated, according to the calibration procedure described in the Materials and Methods section, in order to obtain the number of $^7$Li spins/gram of sample with about 20% error. The result gives $^7$Li spins/gram for Sample $^7$Li-to-$^6$Li as $(0.90 \pm 0.25) \times 10^{19}$ and $^7$Li-control as $(1.22 \pm 0.25) \times 10^{20}$. The one order of magnitude less $^7$Li content in Sample $^7$Li-to-$^6$Li suggests that some of the $^7$Li in the original SEI was removed after the exchange. The quantification of $^6$Li, on the other hand, turned out to be more challenging, although $^6$Li was clearly detected in Sample $^7$Li-to-$^6$Li, as shown in Figure 2b. The small electric quadrupole moment ($Q_{6Li} \approx Q_{7Li}/50$), the small



$^6$Li dipolar linewidths and the quadrupole coupling constant, compared to those for $^7$Li, led to very long relaxation times (typically >2000 s in LiF) which required correspondingly long recycle delays, placing a burden on the stability of the experimental conditions during the course of the extended signal averaging. Nonetheless, $^6$Li spins/gram for $^6$Li-control was $(2.89 \pm 0.78) \times 10^{20}$ with an error of about 27%.

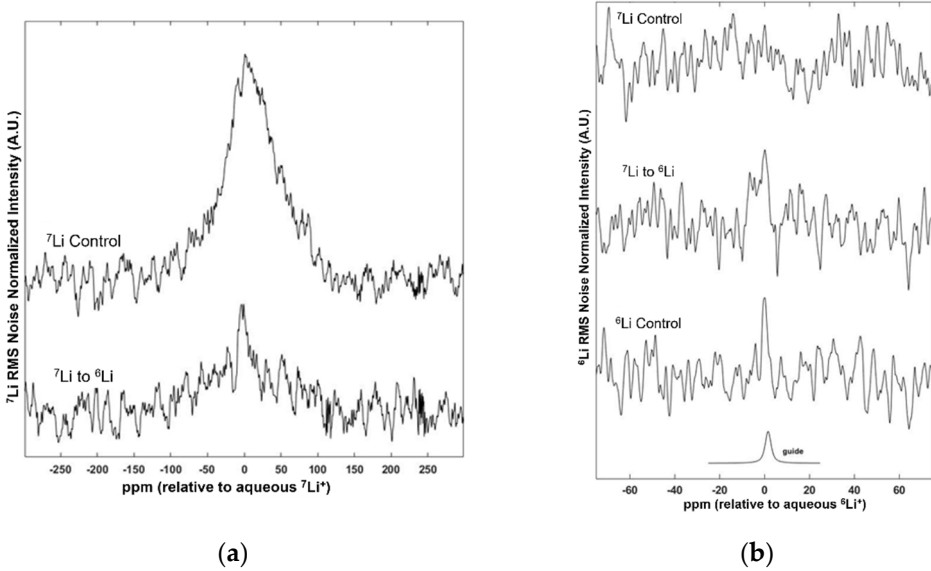

(**a**)  (**b**)

**Figure 2.** Lithium (Li) spectra normalized by RMS noise for delithiated $^7$Li-control, $^6$Li-control, and Sample $^7$Li-to-$^6$Li): (**a**) $^7$Li signal for Sample $^7$Li-to-$^6$Li (430 scans) and $^7$Li-control (324 scans); (**b**) $^6$Li signal for $^6$Li-control (516 scans), Sample $^7$Li-to-$^6$Li (136 scans), and $^7$Li-control (40 scans).

### 2.3. Impact on Capacity Loss

Comparisons of the lithiation/delithiation capacity and capacity loss as a function of cycle number before and after isotope exchange are shown in Figure 3. As illustrated in Figure 3a, a significant amount of capacity was consumed in the first lithiation/delithiation cycle before the exchange. In contrast, the first cycle after the exchange showed a much smaller amount of capacity loss. Further cycling, both before and after exchange, exhibited a gradual decrease in capacity loss, as shown in Figure 3b, although the amount after exchange was less than that before exchange.

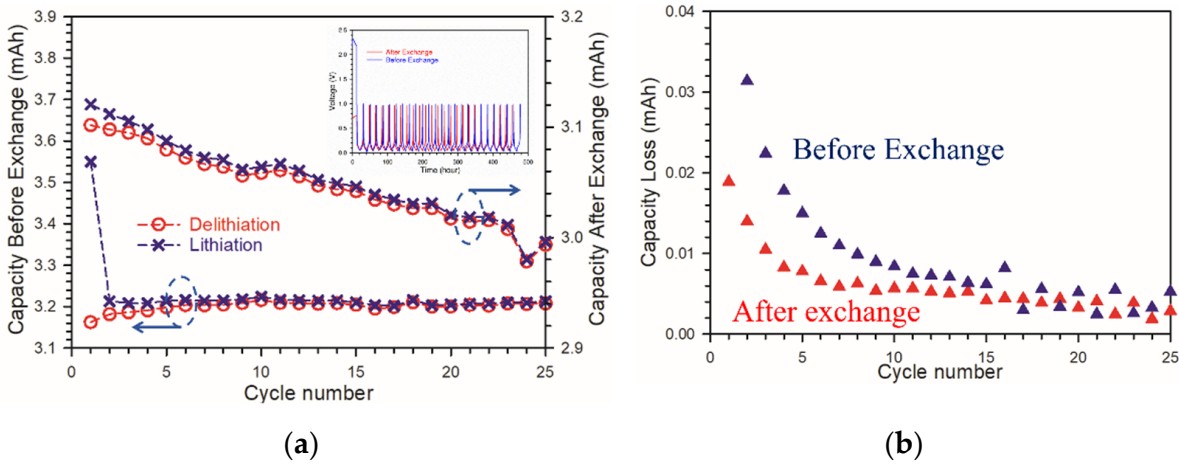

(**a**)  (**b**)

**Figure 3.** (**a**) Lithiation/delithiation capacity as a function of cycle before and after isotopic exchange (inset: cycling voltage profile), (**b**) Cumulative capacity loss as a function of cycle before and after isotopic exchange.

## 3. Discussion

The SIMS analysis showed close to 90% of the $^7Li^+$ being replaced by $^6Li^+$ after one cycle of lithiation/delithiation. If all the exchange was solely driven electrochemically, one would expect the capacity loss before and after exchange to be similar, especially in the first lithiation/delithiation cycle. However, the capacity data do not seem to support that, as indicated by the significant difference in capacity loss in the first lithiation/delithiation cycle between the before and after exchange. This suggests that a portion of the original SEI in the delithiated anode was intact, despite the potential damage caused by the disassembling and rinsing of the anode prior to the exchange. As such, we reasoned that the portion of the isotope exchange observed in the SIMS analysis was a result of chemical diffusion driven by concentration gradient in the absence of applied potential, since a 12-hour resting period was used in the cycling protocol to ensure wetting of the separator. This reasoning is supported by the recent work by Berthault, et al., [23] who found that soaking time of 60 min achieved a complete isotopic exchange.

## 4. Materials and Methods

The electrolyte studied in the present work was 1 molal isotope-enriched lithium bis(trifluoromethanesulfonyl)imide (LiTFSI) in 50/50 (wt%) ethylene carbonate (EC)/ethyl methyl carbonate (EMC). $^6Li$-enriched LiTFSI and $^7Li$-enriched LiTFSI were prepared by reacting $^6Li$ chunks (95 atom% $^6Li$, Sigma Aldrich) and $^7Li$ chunks ($\geq$98 atom% $^7Li$, Sigma–Aldrich) with de-ionized water to form the respective isotopic lithium hydroxide (LiOH). Bis(trifluoromethanesulfonyl)imide acid (HTFSI) (99.0%, TCI Chemicals), which is a solid, was then added slowly to the LiOH until a neutral pH was obtained. The solution was dried until all water was removed. The product was purified via Soxhlet extraction with ethanol, which was then removed by slow heating under a vacuum. EC and EMC were purchased from Gotion and were used as-is. The water content for both solvents was about 20 ppm, according to the product data sheet. The electrolyte solutions were prepared inside an argon-filled glovebox (water and oxygen level $\leq$1 ppm).

Li/Graphite half cells in CR 2032 coin cells with an electrode area of 1.60 cm$^2$ for both the graphite and Li were assembled inside an argon-filled glovebox. The composition of the graphite electrodes (CAMP facility at Argonne National Laboratory) used for the study consisted of 91.83 wt% superior graphite SLC1520P and 2 wt% TIMCAL C45 carbon, with 6 wt% of Kureha 9300 PVDF binder and 0.17 wt% of oxalic acid. The graphite loading was 6.33 mg cm$^{-2}$ on a 10-μm thickness copper foil. Isotope-enriched $^{6,7}Li$ foil of 0.5 mm in thickness was prepared by pressing the isotopic Li chunks into the desired thickness. Celgard CG3501 was used for the separator.

The SEI formation was carried out at 25 °C galvanostatically at a rate of °C (0.2 mA cm$^{-2}$) for Group 1 for 1 cycle, by discharging the cell from OCV to 0.01 V and then re-charging to 2.0 V, followed by a holding period at 2.0 V until the cells were ready to be disassembled, and Group 2 for 25 cycles, discharging the cell from OCV to 0.01 V and then re-charging to 1.0 V, followed by a holding period at 1.0 V until the cells were ready to be disassembled. The cycling procedure was performed on a Maccor series 4000 cycler. After cycling, the coin cells were brought back to the argon-filled glovebox for disassembling, and the graphite anodes were rinsed with EMC three times and then dried under vacuum before isotopic analysis. For the ones that were for isotope exchange, they were reassembled with the other isotope-enriched foil and electrolyte for SEI formation. The disassembly and rinse procedure was deemed necessary to remove background signals from residual electrolyte salt, although it is acknowledged that some portions of the SEI could have been lost in the process.

Isotopic abundance measurements on the graphite anodes were performed using time-of-flight secondary ion mass spectroscopy (ToF-SIMS) for depth profiling, and solid-state NMR for bulk analysis. The samples for the ToF-SIMS study were used as-is, while for the solid-state NMR, the graphite powder was scraped off from the current collector, and about 50 mg of graphite particles were collected to achieve good signal-to-noise level. The

ToF-SIMS measurements were performed at Pacific Northwest National Laboratory using a TOF.SIMS5 instrument (IONTOF GmbH, Münster, Germany) in a dual beam interlaced mode for depth profiling. A 2.0 keV $O_2^+$ beam was used as the sputtering beam and a 50 keV $Bi_3^{2+}$ beam was used as the analysis beam for signal collection. The $O_2^+$ sputtering beam was scanned over a $350 \times 350$ $\mu m^2$ area, and the equivalent sputter rate was about 0.5 nm/s based on the amorphous carbon regularly used for SEM coating. The $Bi_3^{2+}$ beam was focused to about 5 $\mu m$ diameter. The beam current varied from 0.1 to 0.30 pA at a 10 kHz frequency from one measurement to another. The range of $Bi_3^{2+}$ beam current was chosen to ensure reasonably strong $Li^+$ signal intensity and to avoid any uncorrectable signal saturation. The $Bi_3^{2+}$ beam was scanned over an area of $100 \times 100$ $\mu m^2$ at the $O_2^+$ sputter crater center during data collection.

$^7Li$ and $^6Li$ NMR measurements were performed at 117 MHz and 44 MHz, respectively (7T), using a Varian/Agilent DDR solid state NMR spectrometer and a Chemagnetics 3.2 mm MAS probe at Hunter College. Single-pulse direct polarization signals were obtained under ambient conditions with spinning rates of 18–20 kHz and adequate radiofrequency pulse power (pulse nutation rate = 83 kHz). The likelihood that LiF was present in the samples warranted the use of long recycle delays for detection. Therefore, in order to minimize saturation effects, recycle delays of 17 min for $^7Li$ and 35 min for $^6Li$ were allowed between scans. Other acquisition and processing parameters (receiver gain, digitizer resolution, filters, etc.) were held constant. Under these experimental conditions, accumulation and averaging of about a few hundred scans produced manageable spectral signal-to-noise levels. All spectra were referenced to $^6Li$ and $^7Li$ signals in an aqueous LiCl solution. To obtain the $^{6,7}Li$ content in the anode samples, the spectral intensities were measured identically, and calibrated with respect to anhydrous LiF (Sigma–Aldrich, used as the reference standard). In this quantitative procedure, $^{6,7}Li$ NMR measurements were made for LiF references with different masses (up to 75 mg). Each reference spectrum was normalized according to its particular RMS noise and scan count. Following this, integrated spectral intensities were correlated with the (natural abundance) $^{6,7}Li$ content. The obtained results were averaged to provide a calibration for NMR signal intensity per $^{6,7}Li$ spin. The $^{6,7}Li$ content per gram in the anode materials was then obtained by dividing the respective $^{6,7}Li$ integrated signal intensity by the calibration and sample mass.

## 5. Conclusions

Labeling SEI with $^7Li$ and $^6Li$ isotopes and applying solid-NMR and SIMS analyses, our study shows that the $Li^+$ "immobilized" in the chemical ingredients of SEIs are in fact ion-exchangeable at very fast rates, even in the absence of any driving force from an applied electric field. In an actual battery environment, where the $Li^+$ migration gets additional acceleration from the difference between electrochemical potentials between electrodes, $Li^+$ is expected to move faster. This discovery directly conflicts with the known low ion conductivity of the SEI ingredients in bulk state, and strongly implies to us that the ion transport mechanism across SEI significantly differs from that known in liquid or solid bulk. The future investigation of SEIs should focus on this paradox, because its understanding will lay the foundation for the design of new electrolyte materials and interphasial chemistries.

**Author Contributions:** Conceptualization, J.S.H. and K.X.; methodology, J.S.H., Z.Z., P.S., S.G.G., and S.S.Z.; formal analysis, J.S.H., Z.Z., P.S., S.G.G. and S.S.Z.; investigation, J.S.H., Z.Z., S.G.G. and S.S.Z.; data curation, J.S.H., Z.Z., P.S., S.G.G. and S.S.Z.; writing—original draft preparation, J.S.H.; writing—review and editing, J.S.H., Z.Z., P.S., S.G.G., S.S.Z. and K.X. All authors have read and agreed to the published version of the manuscript.

**Funding:** This work was supported by the US Army Department and the Joint Center for Energy Storage Research (JCESR), an Energy Innovation Hub funded by the Department of Energy, Basic Energy Science, under an Interagency Agreement No. IAA SN202095. The SIMS measurements were funded by a project award (10.46936/staf.proj.2020.51724/60000276) from the Environmental Molecular Sciences Laboratory, which is a DOE Office of Science User Facility sponsored by the Biological and Environmental Research program under Contract No. DE-AC05-76RL01830. The NMR measurements performed at Hunter College were funded by the Army Research Laboratory, grant number W911NF-21-2-0221.

**Data Availability Statement:** All data is available in this manuscript.

**Acknowledgments:** The authors are grateful for insightful discussion from colleagues Marshall Schroeder and Jeffrey Read at the Army Research Laboratory, and Bernhard Roling at the University of Marburg, Germany.

**Conflicts of Interest:** The authors declare no conflict of interest.

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
