# Peer review of "Quantifying Lithium Ion Exchange in Solid Electrolyte Interphase (SEI) on Graphite Anode Surfaces"

_inorganics, doi:10.3390/inorganics10050064_

Round 1
Reviewer 1 Report
The authors present a simple study to demonstrate that Li in the SEI is quickly exchanged during cycling, indicating that Li contained in the SEI is still mobile within the system. Generally the study is presented well and the conclusions are kept simple. A few comments to address before publication:
1) the authors say that after 1 cycle 91% of Li is exchanged and after 25 the number is 95%. the authors should comment on the size of error in these measurements - are these values significantly different given all of the potential errors in the measurements?
2) the authors mention SEI damage in the main text - from experience taking apart coin cells can be a relatively damaging act. did the authors characterize the electrode after taking it apart to verify at least that it is not too substantially different in the reassembled cell?
3) I don't understand the difference in capacity loss between before and after exchange - why would electrochemical vs chemical exchange have anything to do with capacity loss. If it is exchanged the total amount of Li in the system is the same? Authors should explain more on their theory here.
Author Response
Reviewer #1
The authors present a simple study to demonstrate that Li in the SEI is quickly exchanged during cycling, indicating that Li contained in the SEI is still mobile within the system. Generally the study is presented well and the conclusions are kept simple. A few comments to address before publication:
Thanks for the positive feedback.
- the authors say that after 1 cycle 91% of Li is exchanged and after 25 the number is 95%. the authors should comment on the size of error in these measurements - are these values significantly different given all of the potential errors in the measurements?
Thanks for the comment. The authors did not mean to imply any statistical significance between the two data points but merely pointed out the observation. The main point of the results was to show that 1 formation cycle was all it took to replace majority of the original SEI. The sentences have been revised.
- the authors mention SEI damage in the main text - from experience taking apart coin cells can be a relatively damaging act. did the authors characterize the electrode after taking it apart to verify at least that it is not too substantially different in the reassembled cell?
Good question and thanks for asking it. This is a long-standing issue in ex-situ analysis. There is always the strong possibility that some of the “evidence” has been destroyed upon cell disassembly and solvent rinsing. Unfortunately for both TOFSIMS and ex-situ NMR, these preparation steps are necessary. We have added this caveat to the manuscript.
- I don't understand the difference in capacity loss between before and after exchange - why would electrochemical vs chemical exchange have anything to do with capacity loss. If it is exchanged the total amount of Li in the system is the same? Authors should explain more on their theory here.
Thanks for pointing out. The results and discussion sections have been revised to clarify the confusion.
Reviewer 2 Report
General comment: This manuscript reported that “Quantifying Lithium Ion Exchange in Solid Electrolyte Inter- 2 phase (SEI) on Graphite Anode Surfaces”. This work attempts to discuss about the formation of SEIs and then quantifying the distribution of permanently immobilized and ion-exchangeable Li+ with solid-state NMR and ToF-SIMS. The present form of this manuscript is not suitable for publication. Therefore, I suggest this manuscript can be accepted after major revisions.
Comment 1: The authors should improve their literature review by adding more recent references
Comment 2: It is better to add the FE-SEM images, EDS, elemental analysis data of the graphite anode before and after cycling.
Comment 3: The authors should add the conclusion part.
Comment 4. The abstract should be more concise and clear.
Comment 5. The results and discussions part is also not well written. It should be improved.
Author Response
Reviewer #2
General comment: This manuscript reported that “Quantifying Lithium Ion Exchange in Solid Electrolyte Inter- 2 phase (SEI) on Graphite Anode Surfaces”. This work attempts to discuss about the formation of SEIs and then quantifying the distribution of permanently immobilized and ion-exchangeable Li+ with solid-state NMR and ToF-SIMS. The present form of this manuscript is not suitable for publication. Therefore, I suggest this manuscript can be accepted after major revisions.
Comment 1: The authors should improve their literature review by adding more recent references
We respectfully disagree on this matter. There are several recent references (2019 – 2021) that demonstrate that our literature review is up to date. However the SEI models presented in the older references are still regarded as relevant and are still prominently cited in current papers on this subject.
Comment 2: It is better to add the FE-SEM images, EDS, elemental analysis data of the graphite anode before and after cycling.
We agree that more analytical work along these lines would add value to this investigation, but they are beyond the scope of our investigation. We have added a sentence about the value of additional possible methods to the text
Comment 3: The authors should add the conclusion part.
We have added a conclusion section.
Comment 4. The abstract should be more concise and clear.
We have edited the abstract accordingly
Comment 5. The results and discussions part is also not well written. It should be improved.
We have made some changes for enhanced clarity.
Reviewer 3 Report
The authors investigated the ion exchange of Li ions in SEI. The manuscript is good for publication in this journal due to its novelty and originality. Minor modifications might help to increase the impact of the manuscript. This reviewer recommends adding the comparison study of additive-induced SEIs and the baseline electrolyte-induced SEI. It will give more insight to the potential reviewers, convincing the experimental methodology. FEC or VC might be an interesting candidate because they have been well known, and the ion conduction mechanisms in the FEC or VC-induced SEI might be different from the baseline due to the polymeric phase, 'polyVC'.
Author Response
Reviewer #3
The authors investigated the ion exchange of Li ions in SEI. The manuscript is good for publication in this journal due to its novelty and originality. Minor modifications might help to increase the impact of the manuscript. This reviewer recommends adding the comparison study of additive-induced SEIs and the baseline electrolyte-induced SEI. It will give more insight to the potential reviewers, convincing the experimental methodology. FEC or VC might be an interesting candidate because they have been well known, and the ion conduction mechanisms in the FEC or VC-induced SEI might be different from the baseline due to the polymeric phase, 'polyVC'
Thanks for the positive feedback. Although the comparison with cells containing the electrolyte additives is a very useful suggestion, we are not in a position to perform additional time-consuming experiments. Instead, we have incorporated this suggestion at the end of the manuscript as a guide to future work in this area.
Round 2
Reviewer 2 Report
Can be accepted in its present form